# Fast Diffusion of the Unassembled PetC1-GFP Protein in the Cyanobacterial Thylakoid Membrane

**DOI:** 10.3390/life11010015

**Published:** 2020-12-29

**Authors:** Radek Kaňa, Gábor Steinbach, Roman Sobotka, György Vámosi, Josef Komenda

**Affiliations:** 1Center ALGATECH, Institute of Microbiology of the Czech Academy of Sciences, 37901 Třeboň, Czech Republic; sobotka@alga.cz (R.S.); komenda@alga.cz (J.K.); 2Institute of Biophysics, Biological Research Center, 6726 Szeged, Hungary; steinbach.gabor@brc.hu; 3Department of Biophysics and Cell Biology, Faculty of Medicine, University of Debrecen, 4032 Debrecen, Hungary; vamosig@med.unideb.hu

**Keywords:** proteins mobility, photosynthesis, FCS, thylakoids, cyanobacteria

## Abstract

Biological membranes were originally described as a fluid mosaic with uniform distribution of proteins and lipids. Later, heterogeneous membrane areas were found in many membrane systems including cyanobacterial thylakoids. In fact, cyanobacterial pigment–protein complexes (photosystems, phycobilisomes) form a heterogeneous mosaic of thylakoid membrane microdomains (MDs) restricting protein mobility. The trafficking of membrane proteins is one of the key factors for long-term survival under stress conditions, for instance during exposure to photoinhibitory light conditions. However, the mobility of unbound ‘free’ proteins in thylakoid membrane is poorly characterized. In this work, we assessed the maximal diffusional ability of a small, unbound thylakoid membrane protein by semi-single molecule FCS (fluorescence correlation spectroscopy) method in the cyanobacterium *Synechocystis* sp. PCC6803. We utilized a GFP-tagged variant of the cytochrome b_6_f subunit PetC1 (PetC1-GFP), which was not assembled in the b_6_f complex due to the presence of the tag. Subsequent FCS measurements have identified a very fast diffusion of the PetC1-GFP protein in the thylakoid membrane (D = 0.14 − 2.95 µm^2^s^−1^). This means that the mobility of PetC1-GFP was comparable with that of free lipids and was 50–500 times higher in comparison to the mobility of proteins (e.g., IsiA, LHCII—light-harvesting complexes of PSII) naturally associated with larger thylakoid membrane complexes like photosystems. Our results thus demonstrate the ability of free thylakoid-membrane proteins to move very fast, revealing the crucial role of protein–protein interactions in the mobility restrictions for large thylakoid protein complexes.

## 1. Introduction

Photochemical reactions in thylakoid membranes (TM) of oxygenic photoautotrophs are driven by light-induced charge separation within Photosystem II (PSII) and Photosystem I (PSI) complexes. The resulting electron flow is mediated by small individual proteins (such as plastocyanin or cytochrome *c*) as well as protein complexes such as cytochrome b6f complex (cyt b_6_f). The last bigger complex, ATP synthase, then catalyzes proton translocation across the membrane. TM is thus a compartment highly crowded with proteins and their complexes that all need to be precisely co-localized with their interaction partners to fulfill their functions [1]. The primary photochemical reactions (light-harvesting, charge separation) require close nano-scale interactions of photosystems with the antenna proteins (e.g., cyanobacterial Phycobilisomes (PBS)) for the efficient flow of excitations [2]. The subsequent electron transport requires an optimal arrangement of all protein components forming the electron transport chain [3]. In this respect, the association of the immobile protein complexes into nano-scale large supercomplexes (nanodomains) is one from the possible strategies in thylakoids [4,5]. These nanodomains can be isolated and characterized in vitro for example isolated nanodomains of FtsH proteases [6], respiratory complexes [7], PSI [8], the PSI-PSII-PBS supercomplex [2], the CurT protein [9], and PratA [10]. Despite the small size of these nanodomains (up to 0.2 µm in diameter), which is close to the resolution limit of standard microscopes, some of them can also be visualized either in isolated thylakoids by AFM microscopy [5] while others also by confocal microscopy in native cells (see, e.g., [6,7,8]). These small nanodomains are typical by the fluorescent signal of the single tagged or autofluorescent protein and they have been identified in various cyanobacterial strains, including ancient *Gloeobacter* [11] and the most common cyanobacterial model organisms *Synechococcus* sp. PCC7942 [7] and *Synechococystis* sp. PCC6803 (hereafter referred to as *Synechocystis*) (see, e.g., [9]).

Recently, we have described the organization of *Synechocystis* pigment-protein complexes into larger membrane areas (with sizes between 0.2 and 1.5 µm) called photosynthetic microdomains (MDs) [12]. In contrast to the nanodomains defined by the presence of a single protein, MDs can be identified by co-localization of three pigment-protein complexes—Photosystems (PSI, PSII) and PBS [12,13,14,15]. It has been identified in *Synechocystis* cells that there are three main types of MDs that differ in PS/PSII/PBS ratios and their localization is surprisingly stable within minutes [12]. In fact, MD organization adapts only very slowly to a changing light environment by the reorganization of Photosystems (especially PSI) in the course of a few days [15]. These special membrane areas are present in TM during the whole dial cycle of *Synechocystis* regardless of cell cycle stage [15]. Except *Synechocystis,* the organization of TM into large MDs was also identified in *Anabaena sp. PCC 7120* [14]. It has been suggested [12] that MDs represent an evolutionary precursor of the plant thylakoid organization into granal/stromal thylakoids [16]. Plant appressed grana regions with a diameter of 0.2–0.5 µm (reviewed in [17]) are very close in size to the cyanobacterial MDs [12] and both have their distinct characteristic PSI/PSII ratios.

The tight protein co-localization into the smaller nanodomains (see, e.g., [6,7,8]) or larger MDs ([12,14]) makes the TM very rich in protein content. TM is therefore characterized by high macromolecular crowding [18] and belongs to the most crowded biological membranes known, the only other comparable in this respect being mitochondrial inner membranes [19]. The complex system of protein co-localization/separation into the nanodomains/microdomains requires strong and abundant protein–protein interactions among the proteins involved. These protein–protein interactions [20] together with high molecular crowding [18,20] and restrictions due to the proposed “borders” of microdomains [12] represent the three main factors possibly restricting free protein diffusion in TMs [3,21,22]. We still do not know whether the mobility of TM proteins is permanently reduced due to the above mentioned restrictions, or some proteins can be (temporally or permanently) “mobilized” at certain conditions. Indeed, such a “mobilization” of PSII has been for instance shown at super high-light conditions [23,24]. Further, it is also an open question how the process of protein assembly into bigger protein-complexes (governed by protein–protein interactions) affects the observed mobility of the single, assembled/unassembled protein. One would expect the mobility to be much higher for unassembled proteins, as it has been indicated by faster diffusion of the free IsiA proteins [25]. However, a possible role of protein assembly into larger complexes in restricting their mobility has not been experimentally tested in detail yet.

In the present study we utilized the fluorescence correlation spectroscopy (FCS) method to determine the diffusion of an unassembled PetC1-GFP fusion protein [26] within *Synechocystis* thylakoids. The FCS method represents a semi-single molecule approach as the method is able to detect very fast protein diffusion while the protein is passing through the very small focal volume of ~1 fL [27,28]. The FCS method identified two distinctly moving fractions of PetC1-GFP that diffused much faster than any previously characterized TM protein from cyanobacteria; the calculated diffusion coefficient was comparable to that of TM lipids [29,30]. The results suggested that, despite the high molecular crowding, the diffusion of unassembled proteins in TM is much less limited than that of their variants stably assembled within the complexes. Our data support the importance of protein–protein interactions in membrane protein mobility.

## 2. Materials and Methods

### 2.1. Strain Preparation

The *Synechocystis* sp. PCC 6803 (hereafter referred to as *Synechocystis*) glucose tolerant sub-strain GT-P was used as a control wild-type (WT) strain [31]. To prepare the PetC1-GFP expressing strain in the same GT-P genetic background we amplified the whole *petC1-GFP* construct from the original *Synechocystis* PetC1-GFP strain [26] that we obtained from Prof. Colin Robinson (University of Warwick, UK). The resulting PCR product contained the *petC1-GFP* gene placed under constitutive *psbAII* promoter, apramycin resistance cassette and ~300 bp up- and down-stream regions to replace the *psbAII* gene. This PCR fragment was transformed into WT (GT-P), the transformed cells were selected on a BG-11 plate containing kanamycin and then fully segregated by an increasing concentration of this antibiotic. The strains were grown photoautotrophically in liquid BG-11 medium on a rotary shaker at 28 °C and under a moderate irradiance of 40 µmol photons m^−2^ s^−1^ given by white fluorescent tubes. Cultures started from agar were imaged in the exponential phase, 2 days after inoculation.

### 2.2. Analyses of Protein Complexes

Isolated membrane protein fraction was prepared as described previously [32]. Cells from exponential growth phase were pelleted, washed, and re-suspended in buffer B (25 mM MES/NaOH, pH 6.5, 10 mM CaCl_2_, 10 mM MgCl_2_, 25% glycerol (*v*/*v*)). They were broken mechanically in Mini-Beadbeater (BioSpec, USA) using balotina beads. The membrane and soluble protein fractions were separated by centrifugation (35,000× *g*, 20 min). The pelleted membranes were washed, and re-suspended in buffer B. For native electrophoresis, membranes (200–400 µg CHL/mL) were solubilized in 1% β-dodecyl-D-maltopyranoside (*w*/*v*), centrifuged (35,000× *g*, 10 min) and supernatant was analyzed in a 4–12% polyacrylamide gel using clear native electrophoresis (CN—PAGE) in the 1st dimension [33]. The gels were color scanned and the CHL fluorescence image was taken using a LAS 4000 camera (Fuji). The individual components of the separated protein complexes were further analyzed for their protein composition in the second dimension using SDS PAGE. CN-PAGE strips were incubated in 62.5 mM Tris/HCl (pH = 6.8) containing 2% (*w*/*v*) SDS and 1% (*w*/*v*) dithiothreitol for 30 min at room temperature and then were placed on the top of a denaturing polyacrylamide gel (12–20%) containing 7 M urea [34]. Separated proteins were stained by SYPRO Orange (Sigma, Germany) and transferred onto a PVDF membrane. Membranes were incubated with primary antibodies specific for PetA (cat. number AS08 306, Agrisera, Sweden), PetC (cat. number AS08 330, Agrisera, Sweden), and GFP (Abcam), and then with secondary antibody conjugated with horseradish peroxidase (Sigma, Germany) that was detected via chemiluminesce signal using the Crescendo substrate (Merck) and was visualized by LAS 4000 camera (Fuji).

### 2.3. Laser Scanning Confocal Microscopy and Fluorescence Correlation Spectroscopy

The images of GFP-PetC1 localization in *Synechocystis* 6803 thylakoids were acquired by an Olympus FV1000 confocal microscope (Olympus, Japan) equipped with an UPLSAPO 100× objective (NA: 1.40, oil immersion) with the following parameters: excitation 488 nm (Ar-ion laser); dichroic mirror: DM405/488/559/635V; detection of GFP/chlorophyll fluorescence between 500–530 nm and 690–790 nm, respectively. Fluorescence correlation spectroscopy (FCS) was performed with an inverted Zeiss 780 (Carl-Zeiss Jena, Germany) confocal microscope using a water immersion objective (C-Apochromat 40×/1.2 W Korr), Ar-ion excitation (488 nm, 135 µW at the objective) and with detection by a GaAsP detector in photon counting mode (498–576 nm). The diffusion of PetC1-GFP was detected on the bottommost TM (face alignment) and correlation functions were analysed by Fluctuation Analyser 4G v14.3.14. software [35] using a 2D model (two-component anomalous diffusion with fluorescent protein-like blinking) of diffusion. The correction for GFP fading was applied in the software before the autocorrelogram was calculated.

## 3. Results

### 3.1. GFP Tagging of PetC1 Protein in *Synechocystis* sp. 6803

Under normal conditions, PetC1 represents the main isoform of Rieske iron-sulphur protein [36] incorporated in the cytochrome b_6_f complex (cyt b_6_f) in *Synechocystis* cells. This ~23 kDa protein is anchored within the TM by a single hydrophobic C-terminal segment [37,38]. However, the protein can be replaced by another copy of PetC expressed from an alternative gene *petC2* without dramatically altering the phenotype [39]. Thus, if mutated PetC1 could not incorporate into cyt *b_6_f*, the phenotype of the mutant should not be affected significantly and localization of the unassembled PetC1 could be followed. This was the case for a newly constructed mutant expressing the PetC1-GFP protein instead of original PetC1. The petC1-GFP strain was phenotypically indistinguishable from the control WT strain and the association of PetC1-GFP with the cyt b_6_f complex was analyzed by 2D electrophoresis combined with immunodetection (Figure 1). Importantly, the fused PetC1-GFP protein was only detected as a free protein and no detectable PetC1-GFP co-migrated with cyt b_6_f. This is in contrast to the WT strain, where PetC1 is clearly incorporated into dimeric cyt b_6_f complex and it is only missing in the monomeric cyt b_6_f, which however results from the disruption of cyt b_6_f during electrophoresis. The detection of PetA documented the presence of the full size dimeric form of cyt b_6_f in the mutant supporting our conclusion that the PetC1-GFP is not incorporated into the cyt b_6_f complex but it is present as a free protein in TM. The new protein consists of three parts: (1) N-terminal lumenal part formed by 250 residues; (2) following membrane spanning alpha helix containing 25 hydrophobic residues that keep the protein within thylakoids; and (3) C-terminal stroma-exposed GFP. All these three parts made from PetC1-GFP a stable TM protein.

### 3.2. Localization of PetC1-GFP Protein in the Synechocystis Cell

The cellular localization of the PetC1-GFP protein in *Synechocystis* was estimated by confocal microscopy (Figure 2A). TMs were imaged based on chlorophyll a (CHL) autofluorescence (Figure 2B) and their signal was then compared with the GFP signal of the PetC1-GFP (Figure 2C) that was acquired simultaneously. CHL fluorescence is emitted mostly from PSII, Figure 2B thus shows heterogeneous MD organization of PSII in thylakoids (i.e., less/more intense spots of CHL fluorescence) in line with the recently described TM heterogeneity [12,13,14,24]. The GFP signal of free PetC1-GFP protein generally overlapped with the CHL emission of PSII (Figure 2A) showing a rather uniform localization of the free PetC1-GFP protein mostly in TM even though PetC1-GFP was not localized in the cyt b_6_f complex (Figure 1). Only a small portion of the PetA1-GFP signal was located more peripherally to CHL forming a narrow “green shell” of the cells, which could reflect the localization of the protein within the plasma membrane (Figure 2).

### 3.3. Measurements of PetC1-GFP Diffusion by FCS

To assess the mobility of the free PetC1-GFP protein in the mutant cells we used the microscopic fluorescence correlation spectroscopy (FCS) method. This semi-single molecule method is suitable to resolve temporal fluctuations in fluorescence intensity caused by repeating events of protein diffusion in the very small focal volume (around 1 fL) of confocal microscopes [43]). We have detected the fluctuations of the GFP fluorescence signal in the “face” or “top position” mode from a single point (Figure 3A) as cells were more stable during measurements and more suitable for the applied 2D model of diffusion than in the side position. Fluorescence signal fluctuations were measured for 30 s (Figure 3A) and showed fluorescence fading that was considered during the data processing (see Materials and Methods). The signal fluctuations were caused by PetC1-GFP passing through the focal volume of microscope. The signal was then correlated in time-domain and the G(τ)—auto-correlation function (ACF) was constructed (Figure 3B). The ACF was then fitted by a 2D diffusion model (see red line in Figure 3B) while fluorescence fluctuations due to the GFP blinking component (~100 µs) were also considered during fitting. To properly fit experimental data, we needed to apply a two-component anomalous diffusion model to minimize residues during fitting (Appendix A). The two components reflecting diffusion differed in the diffusion coefficient (D) and in the parameter of diffusion anomaly described by the alfa exponent (Table 1). The faster fraction of PetC1-GFP (D_1_ = 2.95 µm^2^ s^−1^) had α_1_ = 0.91 indicating sub-diffusion, the slower fraction (D_2_ = 0.145 µm^2^ s^−1^) had α_2_ = 1.31 suggesting super-diffusion of PetC1-GFP. The calculated diffusion coefficient of PetC1-GFP (Figure 3C, Table 1) was much higher (50–500 times) as compared to FRAP measurements of TM proteins of similar size (e.g., IsiA in cyanobacteria [25]; reviewed in [3,21]).

## 4. Discussion

In the present study, we utilized the mutant expressing the GFP-tagged PetC1 protein to assess the mobility of an unassembled membrane protein in the cyanobacterial TM. We found that GFP-tagging of PetC1 inhibits its incorporation into the cyt *b_6_f* complex and this fusion protein accumulates in TMs as a free protein (Figure 2). A possible reason is that this Rieske iron-sulphur protein exhibits a relatively weaker affinity to the complex among the other larger subunits of cyt *b_6_f;* the addition of the tag further decreases this affinity. The dominant localization of PetC1-GFP in TMs is in line with the previous biochemical and microscopic results [26,44]. The PetC1-GFP, unconstrained by protein interactions with cyt *b_6_f*, was utilized as a fluorophore to determine its diffusion coefficient using FCS (Figure 2) [43]. The FCS method requires low concentrations of fluorophores (N < 200 molecules per focal volume, concentration range: 100 pM–1 µM), which was fulfilled in our sample (see Table 1, *n* = 195 for PetC1-GFP). We used the high FCS sensitivity that is allowed by its operation in the photon counting mode. These photons are counted while a few proteins pass through the confocal volume, making FCS a semi-single molecule method [45]. As FCS measures diffusion based on spontaneous fluorescence fluctuations, it makes FCS a more “steady state method” than FRAP relying on displacing the system from equilibrium by photobleaching [45]. It is worth to note that due to concentration limits, the FCS method is hardly applicable for highly abundant autofluorescent CHL-protein complexes like PSII. Therefore, only in a few in vitro studies [46,47,48,49] has the FCS method been used for autofluorescent proteins, mostly for light-harvesting antenna proteins (LHCII) from higher plants thylakoids [47]. Particularly, FCS has been utilized to study the oligomerization state of LHCII in isolated thylakoids [47] or in detergent micelles [48,49]. To our knowledge, our data thus represent one of the first successful in situ FCS application for photosynthetic cells. Our data are comparable with the results obtained with isolated thylakoids from *Chlamydomonas reinhardtii* [47]. Our successful application was made possible by the small size of the cyanobacterial cells (~2 µm, see Figure 2) reducing the effect of scattering and by the low concentration of PetC1-GFP in cells (N~195, see Table 1) as explained above (see, e.g., [43]).

The analysis of FCS data proved the fast diffusion of the unassembled PetC1-GFP (Figure 3, Table 1). We were able to distinguish two fractions of PetC1-GFP with different diffusion coefficients (see Table 1) that differ in their anomalous diffusion described by the alfa exponent. Diffusion anomaly indicates a non-linear relationship between the mean squared displacement (MSD) of proteins in time [50,51]. It shows that proteins in thylakoids do not follow the simple Brownian motion equations for fluids as the totally free movement is limited by the presence of some “border” or an array of corralling proteins [52] that could surround for instance MDs [12]. Besides the anomaly, the diffusion coefficient values were several times higher than expected from previous data obtained by using the FRAP method (see, e.g., reviews [3,21]). We cannot exclude that the fastest fraction (2.95 µm^2^ s^−1^) might be interpreted as diffusion component of PetC1-GFP close to the membrane as suggested for cytoplasmic EGFR-GFP [53] even though similar values were found also for membrane LHCII [47] from isolated thylakoid. On the other hand the lower value (0.14 µm^2^ s^−1^) is very close to that of mitochondrial inner membrane proteins of oxidative phosphorylation OXPHOS [54]. Previous results most appropriate for comparison with our results come from diffusion studies of proteins with a similar size to PetC1-GFP like higher plant LHCII antennae [55] and cyanobacterial IsiA proteins [25]. However, even in these studies the diffusion coefficient of both proteins was about 50 times lower than that of PetC1-GFP (Table 1).

The difference is even more striking when the mobility of unassembled PetC1-GFP (Table 1) is compared with the actual mobility of the whole cyt b_6_f determined in cells of another cyanobacterium *Synechococcus* PCC 7942 using the FRAP method [24]. The discrepancy points to the idea that protein–protein interaction could be an important factor reducing diffusion of TM supercomplexes in various organisms, especially PSII [56,57,58], PSI [24], cyt b_6_f [24] and ATP synthase [24]. The diffusion coefficients of TM proteins calculated from FRAP measurements have often been called an “arbitrary diffusion coefficient” as the other processes, i.e., protein–protein interactions have not been quantified in these models. It is known for some model systems typical for frequent protein–protein interactions (e.g., nuclear proteins [59]) that fluorescence recovery in FRAP is rather slow [28]. Interestingly, the FRAP measurements revealed a similar behavior of large thylakoid photosynthetic protein complexes (see, e.g., PSI, PSII, ATP-synthase, cyt b_6_f mobility in [24]). The FRAP data thus represent a complex intermix of the “free diffusion” and rate of “protein–protein” interaction (binding/unbinding rates) convoluted together [28]. The importance of the interactions needs to be established based on another independent method or on modeling as this has not been done for photosynthetic complexes. Therefore, the fast diffusion of PetC1-GFP determined in this work is not in contradiction with the very slow diffusion of TM proteins detected by FRAP. Our FCS data show a maximal possible diffusion of the unassembled proteins in thylakoid (diffusion capacity) visible when any specific protein–protein interactions are missing or inhibited. This fraction of the fast moving proteins is then conceptually invisible for most of the FRAP setups with longer bleaching periods as the fast diffusion can already occur during less than one second, i.e., during the bleaching period. Therefore, the fastest diffusion components are often “invisible” for conventional FRAP setups [60,61]. On the other hand, FCS is conceptually “blind” to immobilized or slowly moving bound molecules [28] as the duration of a single experiment is often too short (up to 30 s) to detect slow motion (with tens of seconds characteristic time) [61].

The values found for our free PetC1-GFP (D > 0.1 µm^2^ s^−1^, Figure 3) were comparable with fast lipid diffusion (D_lipids_ = 0.3–1.8 µm^2^ s^−1^ [30]) observed in thylakoids. Lipids should be generally less restricted in their diffusion than proteins with the exception of their fraction firmly bound to the TM complexes (see, e.g., [62,63,64]). The bulk lipid majority form a heterogeneous polymorphic mosaic with most of them situated in the lipid bilayer [65,66] enabling fast diffusion (D > 0.1 µm^2^ s^−1^). A similar difference should also be expected for unassembled proteins and those incorporated in the larger complexes by the action of protein–protein interactions. The effect of protein–protein interactions needs to be carefully implemented into any future models of TM protein mobility (see the previous reviews [3,21,22]). Importantly, the high mobility of unassembled proteins could be an important factor in processes like biogenesis/repair and state transitions of photosynthetic membrane complexes. Namely, light sensitive PSII requires frequent repair during which the newly synthesized and initially unassembled D1 protein must replace its old inactive version within the complex ([67]). Based on our measurement showing the ability of fast diffusion of unassembled PetC1-GFP, the new unassembled D1 does not need to be synthesized in the immediate vicinity of the inactive complex due to its putative high diffusion rate. Indeed, the effect of high-light induced protein mobilization has already been proposed for PSII [23]. Similarly, one could expect a similar effect of antenna protein mobilization during state transitions. The process regulates optimal redistribution of excitation energy flow from antennas either to PSI or to PSII [68,69]. In higher plants thylakoids, the redistribution could be allowed by a highly mobile fraction of LHCII antennae, which has been found by FCS in the stromal lamella of *Chlamydomonas reinhardtii* thylakoids (D_LHCII_ = 0.92–2.13 µm^2^ s^−1^ [47]). The change in the LHCII diffusion during state transitions was shown either by FCS [47] or by single particle tracking [70]. We tried to evaluate a change in the mobility of PetC1-GFP in the *Synechocystis* cell during state transitions induced by low blue light (State 1) or by dark adaptation (State 2). We did not detect any reproducible trends in PetC1-GFP diffusion for these two types of sample preparation (*data not shown*). However, we still cannot exclude that state transitions in cyanobacteria could be accompanied by changes in the mobility of the unassembled/free proteins as membrane fluidity affects the process [71]. More experiments are therefore necessary to clarify the proposed link between protein mobility and state transitions in cyanobacteria. New experiments are needed to evaluate a change in the mobility during state transitions for both transmembrane TM proteins (i.e., photosystems) and PBS situated on the TM surface [56]. In fact, it has originally been proposed that PBS mobility is the main mechanism of state transitions in cyanobacteria [72]. However, the concept needs to be revised experimentally by further experiments (see discussion in [69]) due to the presence of PBS blinking [73] that has been shown to affect FRAP measurements of mobility in many fluorescent proteins [74,75] including GFP [76].

## 5. Conclusions

Our FCS data showed that the unassembled TM proteins are able to diffuse very fast. The fraction of the free, unassembled proteins in the membrane (PetC1-GFP with D = 0.14–2.95 µm^2^ s^−1^) is able to cross the small membrane area (e.g., r = 500 nm of granal disk in chloroplast or in cyanobacterial microdomains [12]) in half a second (t = 0.45 s) when it is calculated as unrestricted diffusion time (t = r^2^/4D [3]). The fast diffusion could be crucial at sudden changes in conditions when the fast mobility of a single unassembled protein could allow fast protein trafficking (e.g., between photosynthetic MDs and repair zones). Our data suggest that the movement of the small proteins does not seem to be a limiting factor for efficient photosynthesis.

## Figures and Tables

**Figure 1 life-11-00015-f001:**
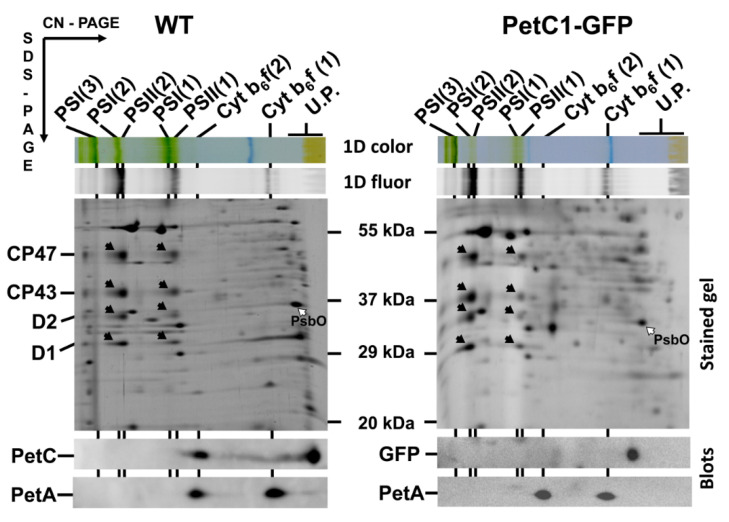
2D analysis of membrane proteins of WT and GFP-PetC1 expressing strains. Membranes isolated from cells were analyzed by 2D CN/SDS-PAGE in combination with immunoblotting. Dimeric and monomeric Cyt b_6_-f complexes were identified using anti PetA antibody, PetC in WT was identified using general anti PetC antibody. Designation of complexes: PSI(3), PSI(2), PSI(1), trimeric, dimeric and monomeric PSI complexes, resp.; PSII(2) and PSII(1), dimeric and monomeric PSII core complexes, resp.; Cyt b_6_-f(2) and Cyt b_6_-f(1), dimeric and monomeric cytochrome b_6_-f complexes, resp.; U.P. unassembled proteins. The large PSII proteins CP47, CP43, D2 and D1 within PSII monomers and dimers (black arrows) and the unassembled PsbO (empty arrow) were identified previously by mass spectrometry [40,41] and by immunoblotting [42]. Each loaded sample contained 5 µg of CHL.

**Figure 2 life-11-00015-f002:**
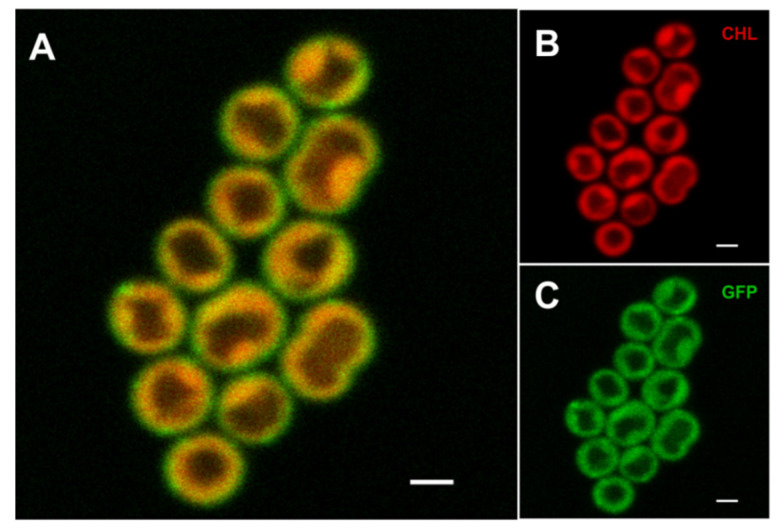
Localization of the free PetC1-GFP fusion protein in *Synechocystis* cells. The combined fluorescence picture (**A**) is formed from the red chlorophyll autofluorescence (**B**)—red channel and from the green GFP fluorescence of PetC1-GFP protein (**Panel C**). Bars represent 1 µm.

**Figure 3 life-11-00015-f003:**
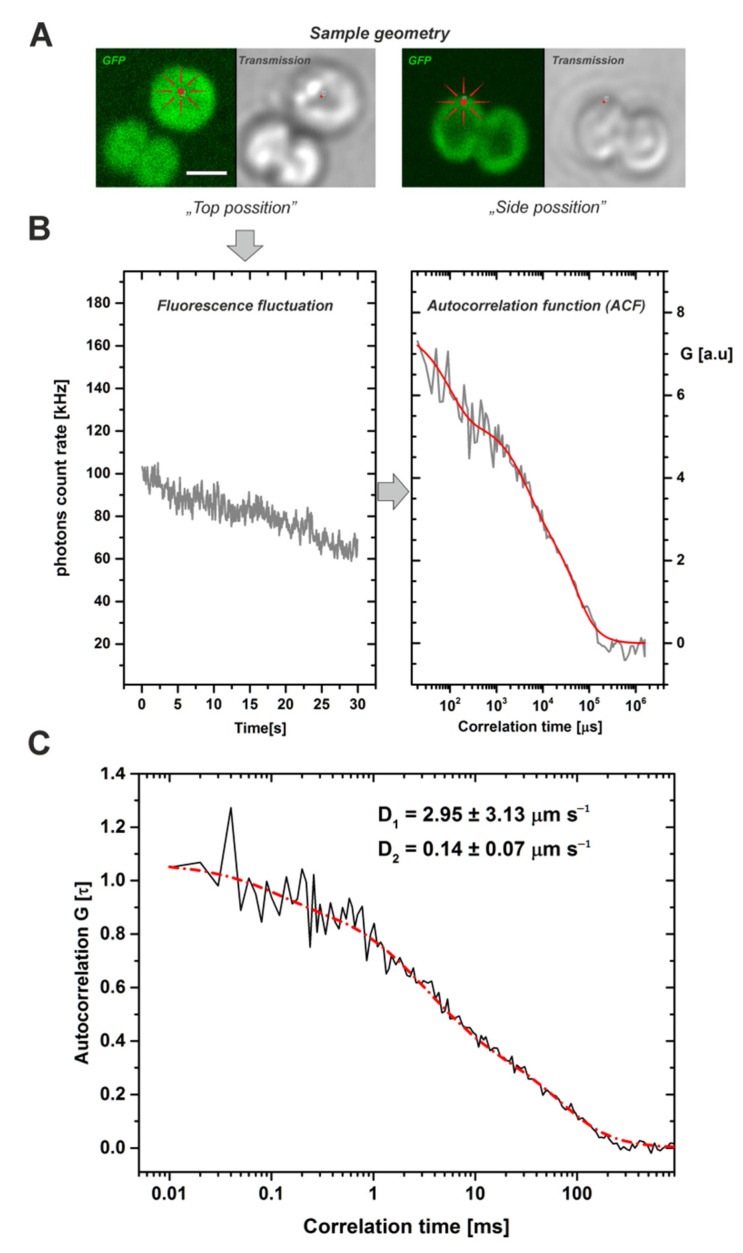
Typical work-flow of FCS measurements leading to calculation of diffusion coefficients for the PetC1-GFP protein in Synechocystis thylakoids. The fluorescence fluctuations of the GFP signal have been detected at a single point, in photon counting mode (detection range 500–530 nm) during 30 s of data acquisition. (**Panel A**): Single cell images of *Synechocystis* thylakoids (GFP fluorescence, transmission) with two different geometrical orientations: (i) top position mode used for the FSC measurements, it means from the face aligned membrane layer (see image on left); (ii) Side position mode (right). Scale bar 2 µm. (**Panel B**): Typical fluorescence fluctuations of GFP signal at a single point of a cell in the top position mode (left). These fluctuations were used after the bleaching correction to construct an auto-correlation function ACF (right). Data represent one single experiment (grey) and the result of the fit (red). (**Panel C**): The averaged cross-correlation functions from measured cells. Black line represents experimental data of calculated autocorrelation functions, red line shows 2D model of diffusion with two diffusion components of PetC1-GFP, D_1_ = 2.954 µm s^−1^, D_2_ = 0.14 µm s^−1^. Number of cells used for analysis was *n* = 79. For more details, see Table 1 and Appendix A.

**Table 1 life-11-00015-t001:** The basic parameters of PetC1-GFP diffusion in thylakoids of *Synechocystis*. The parameters were estimated by fitting the autocorrelation function with a two-component anomalous diffusion model (see Materials and method).

	Fraction I			Fraction II		
N	A_1_	D_1_ [µm^2^ s^−1^]	α_1_	A_2_	D_2_ [µm^2^ s^−1^]	α_2_
195 ± 135	0.73 ± 0.17	2.95 ± 3.13	0.91 ± 0.13	0.27 ± 0.17	0.145 ± 0.071	1.31 ± 0.32

Parameters stand for: N—average number of PetC1-GFP molecules per focal volume; A_1_ and A_2_—relative contribution of fraction I (faster) and fraction II (slower); α—parameter of diffusion anomaly (sub-diffusion for α < 1; super-diffusion for α > 1); D—diffusion co-efficient. Data represent averages and standard deviations for *n* = 79.

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
