# Peer review of "Fast Diffusion of the Unassembled PetC1-GFP Protein in the Cyanobacterial Thylakoid Membrane"

_life, 2020, doi:10.3390/life11010015_

Round 1

Reviewer 1 Report

In the present manuscript, the diffusion of GFP-tagged “free” PetC1 protein in cyanobacterial thylakoid membrane is analyzed by fluorescence correlation spectroscopy. It is first shown that the GFP-tagged PetC1 does not assemble in the cytochrome b6f complex but is instead present as a free protein. The results indicate that free membrane proteins can be highly mobile in the thylakoid membrane in contrast to large protein complexes, which have previously been shown to migrate much slower in the cyanobacterial thylakoid membrane. While I do not think it is suprising that small proteins diffuse faster in the membrane than large protein complexes, I think that the method utilized in the presented study is interesting and could provide further information on the topic in the future. However, I have some concerns and comments on the manuscript presented below:

Major concerns:

1) The aim of the research was to analyze the diffusion of free membrane protein, not proteins that are associated to complexes. Why is the mobility of  PetC1 detected and not some other protein that would naturally be present as a free protein in the membrane? In my opinion it would make more sense to study an individual membrane protein, which actually is a free protein, not a subunit of a complex (even though the  PetC1 is a free protein, when tagged with GFP). 

2) Figure 1. The CN gel as well as the Sypro stained 2D SDS-PAGE look quite different if the WT and PetC1-GFP-lines are compared. First of all, in the PetC1-GFP CN-gel, the PSII dimer band as well as the PSI monomer/PSII monomer bands are much less abundant than in the WT. Is there less of the photosynthetic core complexes in the PetC1-GFP line?

In the 2D-SDS gel, there is a protein spot migrating at around 55 kDa, which is abundant in PetC1-GFP, but is completely absent from WT. What is this protein? Further, in the PetC1-GFP gel there is also another protein spot just below the “D1”-label, which is missing from the WT. What is this spot? It seems that it could be part of Cyt b6f dimer.

Since the diffusion of small membrane protein is studied in the paper, it is important, that the organization of protein complexes in the PetC1-GFP line is similar to WT, because altered protein complex organization would affect the diffusion. Please explain the differences in the 2D-CN/SDS-gels. Also place the protein identification labels so that they do not hide the differences in the 2D SDS-PAGE.

Minor comments:

1) Minor comment on Figure 1.  Are the identified protein spots (the PSII core proteins) analyzed with mass spectrometry or are they obtained from another publication? Please do provide a reference. 

2) Line 43: ATP synthase does not mediate the electron flow between PSII and PSI. But in addition to plastocyanin and cytochrome c, there are several other individual proteins involved in ETC, such as ferredoxin and FNR. Therefore, either write “small individual proteins (such as plastocyanin and cytochrome c) or then list all the important electron carriers.

3) Line 58: should be “-- including ancient Gloeobacter [11] and the most common cyanobacterial model organisms Synechococcus sp. PCC7942—

4) Line 131: I don’t think that Sypro stained gel can be further immunoblotted on pvdf membrane as the proteins should be fixed in the gel prior to Sypro staining and therefore their transfer to the membrane should not be possible. Please explain more clearly.

Author Response

Reply to reviewer 1:

In the present manuscript, the diffusion of GFP-tagged “free” PetC1 protein in cyanobacterial thylakoid membrane is analyzed by fluorescence correlation spectroscopy. It is first shown that the GFP-tagged PetC1 does not assemble in the cytochrome b6f complex but is instead present as a free protein. The results indicate that free membrane proteins can be highly mobile in the thylakoid membrane in contrast to large protein complexes, which have previously been shown to migrate much slower in the cyanobacterial thylakoid membrane. While I do not think it is suprising that small proteins diffuse faster in the membrane than large protein complexes, I think that the method utilized in the presented study is interesting and could provide further information on the topic in the future. 

Response: We are thankful for acknowledging our work.

Point 1: The aim of the research was to analyze the diffusion of free membrane protein, not proteins that are associated to complexes. Why is the mobility of  PetC1 detected and not some other protein that would naturally be present as a free protein in the membrane? In my opinion it would make more sense to study an individual membrane protein, which actually is a free protein, not a subunit of a complex (even though the  PetC1 is a free protein, when tagged with GFP).

Response: We understand the comment. It would be useful to have also another naturally free thylakoid membrane protein that is not assembled in a bigger complex. However it was not our case. On the other hand we managed fluorescently tagged PetC1 by GFP – the new protein PetC1-GFP was then kept unassembled and free in the thylakoid membrane. The PetC1-GFP had 25 hydrophobic residues (forming a strongly hydrophobic helix) that kept the protein in the membrane. Further, the protein had also 250 residues inserted into the lumen and GFP on the stromal part. All the three parts (membrane alfa helix, luminal subunit and stromal GFP) stably anchored the PetC1-GFP in the thylakoid membrane. Therefore, the PetC1-GFP was then very useful for the specific purpose of our study.

Point 2: Figure 1. The CN gel as well as the Sypro stained 2D SDS-PAGE look quite different if the WT and PetC1-GFP-lines are compared. First of all, in the PetC1-GFP CN-gel, the PSII dimer band as well as the PSI monomer/PSII monomer bands are much less abundant than in the WT. Is there less of the photosynthetic core complexes in the PetC1-GFP line?

Response: There is always some variability in the complex pattern in the native gels even when the same sample is analyzed in different native gels and in the particular case of Fig. 1 both samples were analyzed in different native gels. Most often there is variability in the content of various PSI and PSII oligomers. In the analyzed membranes of WT there was a higher content of smaller Photosystem I complexes like dimer and monomer but this does not mean that the overall membrane organization is different and that there are less photosynthetic core complexes. We do not know what is the reason for this variability but always the majority of PSI is present in a trimeric form and this is indeed valid for both analyzed membranes of WT and PetC1-GFP strains. On the other hand, the content of PSII dimer and monomer in both strains is quite similar. It is apparent from chlorophyll fluorescence scan of the native gels which reflects specifically the presence of PSII as all forms of PSI complexes are not sufficiently fluorescent under conditions used (Fig. 1, 1D fluor). We modified the designation of complexes to distinguish PSII and PSI dimers and monomers.

Point 3: In the 2D-SDS gel, there is a protein spot migrating at around 55 kDa, which is abundant in PetC1-GFP, but is completely absent from WT. What is this protein? Further, in the PetC1-GFP gel there is also another protein spot just below the “D1”-label, which is missing from the WT. What is this spot? It seems that it could be part of Cyt b6f dimer.

Response: We do not know what the reviewer means by ‘55 kDa spot abundant in the PetC1-GFP and absent in WT’. At the level of 55 kDa mark there is a strong band belonging to AtpA/B subunits that are mostly present in the complete complex of ATP synthase. This complex is somewhat smaller than the dimer of PSII, a fraction of ATP synthase subunits is also extending to lower molecular weight due to a partial disassembly of the complex. Another strong spot of a similar mass corresponds to a complex of the large RUBISCO subunits, migrating a bit slower in the first dimension than PSI/PSII monomer. These proteins as well as other proteins including main PSII component designated by arrows were identified previously by mass spectrometry in Komenda et al. 2004. This reference was added to the manuscript. The spot below the “D1” label belongs to SbtA, a bicarbonate transporter, which might be induced in autotrophic cultures that are shaken but not bubbled with air or CO2-enriched air. This SbtA protein band is however functionally not related to Cyt b6f dimer.

Point 4: Since the diffusion of small membrane protein is studied in the paper, it is important, that the organization of protein complexes in the PetC1-GFP line is similar to WT, because altered protein complex organization would affect the diffusion. Please explain the differences in the 2D-CN/SDS-gels. Also place the protein identification labels so that they do not hide the differences in the 2D SDS-PAGE.

Response: As explained above, the differences in the 2D-CN/SDS-gels come mostly from the fact that the samples shows do not originate from the same CN gel and there is no indication that the protein complex organization in the petC1-GFP strain is fundamentally different from WT. We changed the placement of labels on the gel.

Minor comment 1: Minor comment on Figure 1.  Are the identified protein spots (the PSII core proteins) analyzed with mass spectrometry or are they obtained from another publication? Please do provide a reference.

Response: We did not perform the identification of the particular spots in the gel shown in Fig. 1 but their assignment was confirmed many times by mass spectrometry as well as by immunoblotting in various articles. For instance, the mass spectrometry analysis was performed in Komenda et al. 2004 JBCH, while immunodetection in Dobakova et al. 2007, Plant Physiology. The references were added to the legend of the figure.

Minor comment 2: Line 43: ATP synthase does not mediate the electron flow between PSII and PSI. But in addition to plastocyanin and cytochrome c, there are several other individual proteins involved in ETC, such as ferredoxin and FNR. Therefore, either write “small individual proteins (such as plastocyanin and cytochrome c) or then list all the important electron carriers.

Response: Corrected

Minor comment 3 Line 58: should be “-- including ancient Gloeobacter [11] and the most common cyanobacterial model organisms Synechococcus sp. PCC7942—

Response: Corrected

Minor comment 4 Line 131: I don’t think that Sypro stained gel can be further immunoblotted on pvdf membrane as the proteins should be fixed in the gel prior to Sypro staining and therefore their transfer to the membrane should not be possible. Please explain more clearly.

Response: The Sypro Orange staining is fully compatible with the following electrotransfer and immunodetection. The staining is performed using the non-fixed gel in the dark in the buffer solution containing just the diluted stain just for one hour. Then it is photographed and used for immediate transfer. Given the 12-20% polyacrylamide concentration used the diffusion of the proteins within and from the gel during one hour staining is negligible.

Reviewer 2 Report

In this paper, the authors, using advanced microscopy techniques, reveal very fast diffusion of the unassembled PetC1-GFP protein in cyanobacterial cells. The observations are of substantial interest with regard to the mobility of protein subunits before their incorporation into a larger complex. In gerenal, the paper is well written; thus it is worth publishing - but I recommend a careful revision: I advise a more cautious interpretation of the data, and some minor corrections.

1/ My main concern is that the authors - throughout the  manuscript - implicitely assume that although the PetC1-GFP (the Rieske) protein remains unassembled, it gets anchored to the membrane. This ~23 kDa protein, a subunit of the cyt b6/f complex, resides mainly in the lumenal aqueous phase, spans the thylakoid membrane with a single hydrophobic segment and is anchored predominantly by electrostatic interactions (Karnauchov et al. 1997 FEBS Lett; Kurisu et al. 2003 Science - useful references also for the readers of Life). Although the co-localization data, presented by the authors, suggest that these proteins are found in the thylakoids (or at least in their close vicinity), the presented data provide no clue of their anchoring. In other terms, it cannot be ruled out that this protein rather than diffusing in the membrane, moves preferentially in the lumen (or gets easily detached from the membrane). Although the authors consider this possibility (ll. 267-269), they appear to treat the data as diffusion of this protein in the lipid membrane, which may or may not be true.

2/ Corrigenda:

  • l. 28: LHCII - l. 210abbreviation without definition
  • l. 170: from the PSII -- from PSII
  • l. 210: (a) -- (A); (b) -- (B)
  • l. 300: to -- too

Author Response

Reply to reviewer 2:

In this paper, the authors, using advanced microscopy techniques, reveal very fast diffusion of the unassembled PetC1-GFP protein in cyanobacterial cells. The observations are of substantial interest with regard to the mobility of protein subunits before their incorporation into a larger complex. In gerenal, the paper is well written; thus it is worth publishing - but I recommend a careful revision: I advise a more cautious interpretation of the data, and some minor corrections.

Thanks for the comments.

Point 1: My main concern is that the authors - throughout the manuscript - implicitly assume that although the PetC1-GFP (the Rieske) protein remains unassembled, it gets anchored to the membrane. This ~23 kDa protein, a subunit of the cyt b6/f complex, resides mainly in the lumenal aqueous phase, spans the thylakoid membrane with a single hydrophobic segment and is anchored predominantly by electrostatic interactions (Karnauchov et al. 1997 FEBS Lett; Kurisu et al. 2003 Science - useful references also for the readers of Life). Although the co-localization data, presented by the authors, suggest that these proteins are found in the thylakoids (or at least in their close vicinity), the presented data provide no clue of their anchoring. In other terms, it cannot be ruled out that this protein rather than diffusing in the membrane, moves preferentially in the lumen (or gets easily detached from the membrane). Although the authors consider this possibility (ll. 267-269), they appear to treat the data as diffusion of this protein in the lipid membrane, which may or may not be true.

Response: We understand the reviewer’s concern as it could justify the observed fast migration of the protein. However, we consider the possibility mentioned by the reviewer highly improbable. The protein behaves like standard membrane protein; it is co-isolated with membranes as it is anchored in the membrane by a single transmembrane helix of sufficient hydrophobicity and length. The protein is synthesized on the membrane and the first 250 residues are inserted into the lumen but the following 25 hydrophobic residues forms a strongly hydrophobic helix which must remain embedded in the membrane. Furthermore, unlike the native protein, the PetC1-GFP has a GFP attached to the C-terminus which is on the stromal side. GFP protein is quickly folded into its very stable 3D structure which will definitely prevent the transfer of the whole protein into the lumen. We have added few sentences on the point and added the references.

Corrigenda: Corrected